# Health facility readiness to provide antenatal care (ANC) and non-communicable disease (NCD) services in Nepal and Bangladesh: Analysis of facility-based surveys

**Deependra K. Thapa**[1,2]*, **Kiran Acharya**[3], **Anjalina Karki**[1], **Michelle Cleary**[2]

**1** Nepal Public Health Research and Development Center, Kathmandu, Nepal, **2** School of Nursing, Midwifery & Social Sciences, Central Queensland University, Sydney, Australia, **3** New ERA, Rudramati Marga, Kalopul, Kathmandu, Nepal

* thapa.deepen@gmail.com

## Abstract

### Background

Antenatal care (ANC) visits provide an important opportunity for diagnostic, preventive, and curative services for non-communicable diseases (NCDs) during pregnancy. There is an identified need for an integrated, system-wide approach to provide both ANC and NCD services to improve maternal and child health outcomes in the short and long term.

### Objective

This study assessed the readiness of health facilities to provide ANC and NCD services in Nepal and Bangladesh, identified as low–and middle–income countries.

### Method

The study used data from national health facility surveys in Nepal (n = 1565) and Bangladesh (n = 512) assessing recent service provision under the Demographic and Health Survey programs. Using the WHO's service availability and readiness assessment framework, the service readiness index was calculated across four domains: staff and guidelines, equipment, diagnostic, and medicines and commodities. Availability and readiness are presented as frequency and percentages, while factors associated with readiness were examined using binary logistic regression.

### Results

Of the facilities, 71% in Nepal, and 34% in Bangladesh reported offering both ANC and NCD services. The proportion of facilities which showed readiness for providing ANC and NCD services was 24% in Nepal and 16% in Bangladesh. Gaps in readiness were observed in the availability of trained staff, guidelines, basic equipment, diagnostics, and medicines. Facilities managed by the private sector or a Non-Governmental Organization, located in an

**Data Availability Statement:** The datasets used on the current study are available on the Demographic and Health Survey Program repository, which is a

public, open access repository (https://dhsprogram.com/data/available-datasets.cfm).

**Funding:** The author(s) received no specific funding for this work.

**Competing interests:** The authors have declared that no competing interests exist.

urban area, with management systems to support the delivery of quality services were positively associated with readiness to provide both ANC and NCD services.

## Conclusion

There is a need to strengthen the health workforce by ensuring skilled personnel, having policy, guidelines and standards, and that diagnostics, medicines, and commodities are available/provided in health facilities. Management and administrative systems are also required, including supervision and staff training, to enable health services to provide integrated care at an acceptable level of quality.

## Background

Noncommunicable diseases (NCDs) are a leading cause of mortality, morbidity and disability, accounting for more than 41 million deaths (71% of all deaths) globally per year [1]. NCDs were responsible for 1.62 billion Disability Adjusted Life Years (DALYs) in 2019, an increase from 43.2% in 1990 to 63.8% of total DALYs in 2019 [2]. There has been a surge in the burden of NCDs in low- and middle-income countries (LMICs) due to the globalization of unhealthy lifestyles and the increased number and proportion of the ageing population. The burden of NCDs has disproportionately affected LMICs, with more than three quarters of NCD-related deaths occurring in these regions [3, 4]. A large disparity remains between high- and low-income countries regarding strategies to control NCDs, reliable and comprehensive data on risk factors, and surveillance systems [5]. Despite the increasing risk factor exposure, prevalence and mortality related to NCDs in LMICs, public health policy responses have been slow, limited, and inadequate.

Whilst people of all age groups, sexes and regions are affected by NCDs, these conditions are often overlooked, undiagnosed, and untreated among women [6]. NCDs contribute to more than 18 million deaths among women—more than two-thirds of all female deaths each year [7]. The burden of NCDs is also increasing among women of reproductive age, with NCDs becoming a significant cause of female death during childbearing age in LMICs [8]. The enduring myth that health issues among women are primarily related to their reproductive health has undermined recognition of the importance of NCDs affecting women.

Women with health conditions such as hypertension, anaemia, malnutrition, obesity and heart disease are at higher risk of pregnancy- and child-birth-related complications, increasing risks for the mother and her offspring. Almost 28% of maternal deaths around the world are attributed to chronic health conditions exacerbated during pregnancy and childbirth [9]. Children of women with NCDs are also at increased risk of adverse health outcomes, including NCD conditions such as obesity, diabetes, stroke and cardiovascular diseases later in life. The intergenerational impact of NCDs during pregnancy can multiply the continuing NCD pandemic [10]. Mahajan et al. [11] identified several challenges for women in low-income countries, including a lack of awareness of NCDs and associated risk factors. A study in Tanzania, for example, showed almost 80% of reproductive age women were not aware of their hypertensive condition [12].

Interventions designed to improve maternal and child health have primarily focused on the treatment and management of the presenting health condition during pregnancy, delivery, and the postpartum period, with limited attention given to the underlying causes [13]. The epidemiological transition to NCDs, including the burden of preventable maternal morbidity and

mortality, is likely to hamper the achievement of sustainable development goals in LMICs. Recognizing the inextricable link between maternal and child health and NCDs, ensuring a more integrated and holistic approach to prevention and care at the primary health care level, is a requirement, especially in low resource settings. A critical strategy to enhance maternal and child health is to ensure a continuum of care by strengthening health systems and improving the quality of health care [14, 15].

Primary health care in most LMICs encompasses services related to maternal health during pregnancy, delivery, and the postnatal period. Antenatal care (ANC) coverage has increased substantially during the past decade. A study by Tikmani et al. [16] analysing trends of ANC visits in LMICs from 2011 to 2017 showed that almost all women attended an appointment at least once, and there was a significant increase in the proportion of women who made at least four ANC visits. The WHO's updated ANC guideline recommends a minimum of eight antenatal care visits during pregnancy so that the well-being of mothers and newborns is ensured [17].

Whilst the Millennium Development Goals Report suggests significant progress in maternal and child health, there is still high maternal and neonatal mortality, mostly in LMICs, from preventable pregnancy- and birth-related complications [18, 19]. ANC visits provide a critical opportunity to diagnose and manage pregnancy-related complications to improve maternal and child health. In addition, ANC visits also provide opportunity for early screening of modifiable risk-factors and identification of pre-existing conditions [20]. Despite the evidence on the health and economic benefits of integrating maternal health and NCDs, the link between these services remains neglected, largely due to the traditional approaches that divide service delivery into communicable diseases, NCDs, and maternal and child health [10]. Maternal health programs in LMICs are well placed to integrate NCD care programs [21].

The need for an integrated, system-wide approach to ANC and NCDs to improve maternal and child health outcomes is well established. The integration of these services can ensure screening for NCDs, targeted interventions, and support for lifestyle modifications [22] as well as long-term population health. This may also improve maternal and child health outcomes in the short term. Readiness of health systems to provide both ANC and NCD services is important for successful integration of ANC and NCD services. To date, evidence of service readiness for ANC and NCD services is not well documented, with studies examining the readiness of health facilities to provide integrated services lacking.

The present study aimed to explore the readiness of health facilities to provide ANC and NCD services using data from health facilities surveys in two LMICs countries: Nepal and Bangladesh, and assess organizational factors associated with readiness to provide these services. Whilst these two countries have different cultures and practices in relation to healthcare, there are many challenges common to both as LMICs. Information on service readiness to provide ANC and NCD services in these two countries will provide an important insight for policy makers to integrate these services for improving health of women and general population in South Asia and other LMICs. This will also inform the further development of global aims to be achieved in the context of the Sustainable Development Goals (SDGs).

## Methods

### Data source

This study used data from the Service Provision Assessments (SPA) conducted under the USAID's Demographic and Health Surveys Program [23] which includes standardized health facility audit and health service provider interview data. The cross-sectional survey of health facilities provides comprehensive information on the availability and readiness of basic health care services for each country including child health, maternal and newborn care. SPA surveys

**Table 1. Survey year and sample size of health facilities.**

| Country | Survey Year | Total number of surveyed health facilities | Number of health facilities excluded[†] | Sample size for current study (after exclusion) |
|---|---|---|---|---|
| Nepal | 2021 | 1576 | 11 | 1565 |
| Bangladesh | 2017 | 1524 | 1012 | 512 |
| Total | | 3100 | 1023 | 2077 |

[†] Stand-alone HTCs and community clinics were excluded from the analysis because they are not expected to provide ANC and/or NCD services.

provide essential data which enables health system to be monitored and strengthened in LMICs [23].

This study included two LMICs in South Asia: Nepal and Bangladesh, where recent Demographic and Health Program SPA surveys had been conducted. These surveys were the Nepal Health Facility Survey (NHFS) 2021, and the Bangladesh Health Facility Survey (BHFS) 2017. The health facility survey, globally called the SPA, provides nationally representative estimates, collecting information from health facilities managed by the government, non-governmental organizations (NGO) and private for-profit organizations across the country. Details of survey methodology and sampling strategy are available in the published survey reports; The Ministry of Health and population/Nepal et al. [24] for Nepal, and The National Institute of Population Research Training—NIPORT et al. [25] for Bangladesh.

The recent health facility surveys in the selected two countries collected information from 1576 health facilities in Nepal and 1524 in Bangladesh. Stand-alone HIV Testing and Counselling Centres (HTCs) in Nepal and community clinics in Bangladesh were excluded from the current study (Table 1).

## Measurement of variables

**Dependent variable–readiness for integrating ANC and NCD services.** The dependent variable in this study was readiness for providing ANC and NCD services at health facilities, defined as the availability of services and capacity of health facilities to provide both ANC and NCD services. The readiness index was calculated based on the scores of ANC and NCD service readiness. NCD service included any service offering diagnosis and/or management of diabetes, cardiovascular diseases, and chronic respiratory diseases. Based on the WHO Service Availability and Readiness Assessment (SARA) Manual [23], we identified four domains of service readiness for ANC and NCD, which included staff and guidelines, equipment, diagnostic, and medicines and commodities. The different tracer items for each specific domain of ANC and NCD are provided in Table 2.

Service readiness scores for ANC and NCD were calculated by adding the presence of domain-specific indicators, providing equal weight to each domain and each indicator within the domain. As there were four domains in both ANC and NCD, each domain accounted for 25% of the readiness score. The weighting for each indicator within the domain was equal to 25% divided by the number of indicators in the specific domain. The details of the score calculation is provided in Table 2. The scores for ANC and NCD joint service readiness for each facility were calculated by summing the percentages. Health facilities with scores of 50% and above for both ANC and NCD services were considered ready for providing both ANC and NCD services, while those scoring less than 50% for any of the two services were not considered ready [23, 26]. Thus, a binary variable of readiness to provide ANC and NCD services (Yes/No) was considered as the outcome variable for this study.

**Independent variables.** The independent variables for the study were facility managing authority (public or private/NGO), location (rural/urban), routine quality assurance activities

**Table 2. Summary of readiness domains, indicators, and measurement procedure of ANC and NCD service readiness score for health facilities.**

| Domain | Indicators (Tracer items) | Measurement | Percent score (%) | |
| --- | --- | --- | --- | --- |
| | | | Indicator | Domain |
| **ANC service readiness index** | | | | |
| Staff and guidelines | Guidelines for diagnosis and treatment of ANC | Yes | 12.50 | 25.00 |
| | | No | 0.00 | |
| | At least one staff member trained in ANC | Yes | 12.50 | |
| | | No | 0.00 | |
| Equipment | Blood pressure (BP) apparatus | Yes | 25.00 | 25.00 |
| | | No | 0.00 | |
| Diagnostics | Hemoglobin (Hb) | Yes | 12.50 | 25.00 |
| | | No | 0.00 | |
| | Urine dipstick- protein | Yes | 12.50 | |
| | | No | 0.00 | |
| Medicines and commodities | Iron and folic acid combined tablets | Yes | 8.33 | 25.00 |
| | | No | 0.00 | |
| | Tetanus diphtheria vaccine | Yes | 8.33 | |
| | | No | 0.00 | |
| | Albendazole | Yes | 8.33 | |
| | | No | 0.00 | |
| **Total ANC readiness index score** | | | | 100.00 |
| **NCD service readiness index** | | | | |
| Staff and guidelines | Guidelines for diabetes, CVD and CRD diagnosis and treatment | Yes | 12.50 | 25.00 |
| | | No | 0.00 | |
| | At least one staff member trained in diabetes, CVD, and CRD diagnosis and treatment | Yes | 12.50 | |
| | | No | 0.00 | |
| Equipment | Stethoscope | Yes | 3.57 | 25.00 |
| | | No | 0.00 | |
| | Blood pressure apparatus | Yes | 3.57 | |
| | | No | 0.00 | |
| | Adult scale | Yes | 3.57 | |
| | | No | 0.00 | |
| | Measuring tape (height board/stadiometer) | Yes | 3.57 | |
| | | No | 0.00 | |
| | Oxygen | Yes | 3.57 | |
| | | No | 0.00 | |
| | Peak flow meter apparatus | Yes | 3.57 | |
| | | No | 0.00 | |
| | Spacers for inhalers | Yes | 3.57 | |
| | | No | 0.00 | |
| Diagnostics | Blood glucose | Yes | 8.33 | 25.00 |
| | | No | 0.00 | |
| | Urine protein | Yes | 8.33 | |
| | | No | 0.00 | |
| | Urine glucose | Yes | 8.33 | |
| | | No | 0.00 | |

(*Continued*)

**Table 2.** (Continued)

| Domain | Indicators (Tracer items) | Measurement | Percent score (%) | |
|---|---|---|---|---|
| | | | Indicator | Domain |
| **ANC service readiness index** | | | | |
| Medicines and commodities | Metformin cap/tab | Yes | 1.67 | 25.00 |
| | | No | 0.00 | |
| | Glibenclamide cap/tab | Yes | 1.67 | |
| | | No | 0.00 | |
| | Insulin regular injectable | Yes | 1.67 | |
| | | No | 0.00 | |
| | Glucose 50% injectable | Yes | 1.67 | |
| | | No | 0.00 | |
| | Gliclazide tablet or glipizide tablet (only collected in Bangladesh HFS) | Yes | 1.67 | |
| | | No | 0.00 | |
| | ACE inhibitor (e.g., enalapril, lisinopril, ramipril, perindopril) | Yes | 1.67 | |
| | | No | 0.00 | |
| | Hydrochlorothiazide tablet or other thiazide diuretic tablet | Yes | 1.67 | |
| | | No | 0.00 | |
| | Beta blocker (e.g., bisoprolol, metoprolol, carvedilol, atenolol) | Yes | 1.67 | |
| | | No | 0.00 | |
| | Calcium channel blockers (e.g., amlodipine) | Yes | 1.67 | |
| | | No | 0.00 | |
| | Aspirin cap/tab | Yes | 1.67 | |
| | | No | 0.00 | |
| | Salbutamol inhaler | Yes | 1.67 | |
| | | No | 0.00 | |
| | Beclomethasone inhaler | Yes | 1.67 | |
| | | No | 0.00 | |
| | Prednisolone cap/tab | Yes | 1.67 | |
| | | No | 0.00 | |
| | Hydrocortisone injection | Yes | 1.67 | |
| | | No | 0.00 | |
| | Epinephrine injectable | Yes | 1.67 | |
| | | No | 0.00 | |
| **Total NCD readiness index score** | | | | 100.00 |

(performed/not performed), system to obtain client feedback (No/Yes), external supervision (occurred/ did not occur in previous 4 months), and regular monthly managing meetings (No/Yes). Location of the facilities (rural and urban) were classified as the rural areas (rural municipality) and urban areas (metropolitan/sub metropolitan city and municipality). Routine quality assurance activity was categorized as "Performed" for facilities routinely conducting quality assurance activities as documented by report of quality assurance activities, and "Not performed" for those without quality assurance activities. External supervision measured whether health facilities had received external supervision in the past three months or not. The management level independent variable was categorized as "Yes" for facilities that performed monthly management meetings and "No" for facilities which did not conduct such meetings at least monthly. Regular management meetings are not reported in the BHFS, and is not included in the current analysis for Bangladesh. The selection of independent variables was informed by the literature on facility readiness [26–29].

## Data analysis

Most variables reported in this study were categorical, and are summarized using proportions and then presented in a table for each country. The relationship of the outcome variable–readiness for integrating ANC and NCD services–with the defined independent variables was analyzed using binary logistic regression models. Unadjusted and adjusted odds ratio (OR) was used, and the *p*-value and 95% confidence interval (CI) for the odds ratios (OR) was used to measure the significance level. A *p*-value of 0.05 or lower was considered a statistically significant association. Before fitting the model, correlations between independent variables and the outcome variable were checked, with no significant correlations observed, hence all independent variables were included. Data were analyzed using STATA 15.0. The complex sample design used in the SPA surveys was accounted for by using the "svy" command in the STATA software. Sampling weights were used to correct for non-responses and disproportionate sampling.

## Ethical considerations

This study analyzed existing SPA survey data sets that are freely available upon request. SPA survey protocols undergo ethical review by the United States ICF's institutional review board. These surveys also undergo ethical review in their respective countries.

## Results

### Background characteristics of selected health facilities

Table 3 shows the characteristics of health facilities according to the distribution of covariates used in this study, which included 1565 health facilities in Nepal, and 512 in Bangladesh. In Nepal, most health facilities (92.6%) were managed by the public authority, while in Bangladesh 79.2% were managed by public authority. About 79% of health facilities in Bangladesh were located in urban areas, whereas in Nepal, health facilities were evenly split between rural and urban. Around one fifth of health facilities in Nepal and Bangladesh reported undertaking quality assurance activities. About half of the facilities in Nepal (54.1%) and Bangladesh (46.4%) had systems to obtain client feedback. In the past four months, a higher proportion of facilities were supervised externally; 87.9% in Bangladesh, and 66.2% in Nepal. About two thirds (64.0%) of the health facilities in Nepal reported regular monthly management meetings (Table 3).

### Availability and readiness for ANC, NCD and joint readiness for ANC and NCD services

Table 4 presents the distribution of the availability of, and readiness for providing, ANC and NCD services among study health facilities. Most health facilities in Nepal (98%) and Bangladesh (97%) reported offering ANC services, while 72.0% of health facilities in Nepal and 35.1% in Bangladesh reported offering NCD services. After combining these services, 70.9% in Nepal, and 34.1% in Bangladesh reported offering both ANC and NCD services.

There was substantial variation in the readiness for providing ANC and NCD services across the study countries. Health facilities in both countries showed lower scores in staff and guidelines for ANC service (34.8% in Bangladesh, and 18.7% in Nepal). Compared to Bangladesh (33.6%), the readiness of the diagnostics components of the ANC services was lower in Nepal (27.0%). In terms of ANC service readiness, health facilities were in general strong in the equipment domains. A higher proportion of facilities in Nepal were ready in terms of medicines/commodities (70.9%) compared to Bangladesh (59.7%) (Table 4).

**Table 3. Characteristics of health facilities according to covariates.**

| Variable | Nepal n (%) | Bangladesh n (%) |
|---|---|---|
| **Managing Authority** | | |
| Public | 1448 (92.6) | 402 (79.2) |
| Private/NGO | 116 (7.4) | 106 (20.8) |
| **Location** | | |
| Rural | 730 (46.7) | 404 (78.9) |
| Urban | 834 (53.3) | 108 (21.1) |
| **Routine quality assurance** | | |
| Not performed | 1201 (76.7) | 409 (80.0) |
| Performed | 364 (23.3) | 102 (20.0) |
| **System to obtain client feedback** | | |
| No | 718 (45.9) | 274 (53.6) |
| Yes | 847 (54.1) | 237 (46.4) |
| **External supervision in the last 4 months** | | |
| Did not occur | 529 (33.8) | 62 (12.0) |
| Occurred | 1036 (66.2) | 450 (87.9) |
| **Regular monthly management meetings** | | |
| No | 564 (36.0) | - |
| Yes | 1001 (64.0) | - |
| **Total** | 1565 | 512 |

Regarding NCD service readiness, fewer study facilities had readiness in the staff and guidelines domains (5.7% in Bangladesh and 11.1% in Nepal). Around half of the facilities in Nepal (51.5%) and 39.0% in Bangladesh had readiness in the equipment domain. The proportion of facilities showing NCD service readiness in the diagnostics domain was 21.6% in Nepal, and 20.5% in Bangladesh. A higher proportion of facilities in Nepal (29.2%) compared to Bangladesh (7.5%) had readiness in terms of medicines and commodities (Table 4).

**Table 4. Service availability and readiness to provide ANC and NCD services.**

| Indicators | Nepal (N = 1565) | | Bangladesh (N = 512) | |
|---|---|---|---|---|
| **Service availability** | **n** | **% (95% CI)** | **n** | **% (95% CI)** |
| ANC | 1538 | 98.4 (97.7–98.8) | 494 | 96.5 (94.7–97.7) |
| NCD | 1127 | 72.0 (68.7–75.1) | 180 | 35.1 (32.1–38.1) |
| Both ANC and NCD | 1109 | 70.9 (67.6–74.0) | 174 | 34.1 (31.2–37.2) |
| **Domains of ANC readiness** | | | | |
| Staff and guidelines | 293 | 18.7 (16.6–20.8) | 178 | 34.8 (32.4–37.2) |
| Equipment | 1498 | 95.7 (94.3–97.1) | 459 | 89.6 (87.0–92.1) |
| Diagnostics | 423 | 27.0 (24.3–29.7) | 172 | 33.6 (30.8–36.5) |
| Medicine and commodities | 1166 | 74.5 (73.2–75.7) | 306 | 59.7 (58.1–61.3) |
| **Domains of NCD readiness** | | | | |
| Staff and guidelines | 174 | 11.1 (9.4–12.7) | 29 | 5.7 (5.0–6.5) |
| Equipment | 806 | 51.5 (50.3–52.7) | 200 | 39.0 (37.0–41.0) |
| Diagnostics | 338 | 21.6 (19.4–23.9) | 105 | 20.5 (18.0–22.9) |
| Medicines and commodities | 457 | 29.2 (28.2–30.2) | 38 | 7.5 (6.5–8.5) |
| **Overall facility readiness** | | | | |
| ANC readiness | 836 | 53.4 (52.4–54.5) | 278 | 54.4 (53.0–55.9) |
| NCD readiness | 444 | 28.4 (27.4–29.3) | 93 | 18.2 (17.1–19.3) |
| Readiness for both ANC and NCD | 379 | 24.2 (21.4–27.2) | 83 | 16.3 (14.1–18.7) |

**Table 5. Availability and readiness of ANC and NCD services.**

| | Nepal (N = 1565) | | Bangladesh (N = 512) | |
|---|---|---|---|---|
| | Availability (%) | Readiness† (%) | Availability (%) | Readiness† (%) |
| Only ANC | 27.4 | 44.2 | 62.4 | 55.3 |
| Only NCD | 1.1 | 0.5 | 1.0 | 0.0 |
| Both ANC and NCD | 70.9 | 24.2 | 34.1 | 16.3 |
| Neither ANC nor NCD | 0.6 | 31.1 | 2.6 | 28.4 |

†Health facilities with scores of 50% and were considered ready for providing services, while those scoring less than 50% were not considered ready.

Table 4 also shows the overall facility readiness for ANC, NCD and joint readiness for both ANC and NCD services. The overall readiness to offer specific services for ANC and NCD among the study facilities was 53.4% and 28.4% respectively in Nepal, and 54.4% and 18.2% respectively in Bangladesh. In terms of readiness for providing both ANC and NCD services, 24.2% of the study facilities in Nepal, and 16.3% in Bangladesh were ready to provide both services.

Availability and readiness of services were compared across the four groups: only ANC, only NCD, both ANC and NCD, and neither ANC nor NCD (Table 5). Facilities having availability and readiness for only ANC was 27.4% and 44.2% respectively in Nepal, while 62.4% and 55.3% facilities in Bangladesh had only ANC service availability and readiness respectively. Few facilities were providing only NCD service in both countries. The availability and readiness for both ANC and NCD was higher in Nepal, compared to Bangladesh. Nearly one-third of the facilities in Nepal (31.1%) and Bangladesh (28.4%) were not ready to provide ANC or NCD.

Fig 1 shows that NCD service availability and readiness among the facilities offering ANC services in Nepal (n = 1538) were 72.1% and 28.3% respectively, while in the case of Bangladesh (n = 494), NCD service availability and readiness within ANC facilities was 35.4% and 18.5% respectively.

## Factors associated with facility readiness

Table 6 presents the unadjusted and adjusted ORs using binary logistic regression to identify factors associated with facility readiness for both ANC and NCD services. Similar patterns were observed in both unadjusted and adjusted analyses. In bivariate logistic regression analyses, managing authority type, routine quality assurance, availability of system to obtain client feedback, external supervision conducted, and regular monthly management meetings were positively associated with readiness for ANC and NCD services in Nepal. For Bangladesh, all the covariates measured were associated with readiness for providing both services.

In the final adjusted models, some variations in covariates in terms of level of significance and the strength of association were observed across the study countries. Nepal showed that health facilities managed by the private sector or an NGO (aOR = 10.1; 95% CI:6.5–15.7) compared to the public sector; facilities performing routine quality assurance (aOR = 1.9; 95% CI: 1.2–2.8); having a system to obtain client feedback (aOR = 1.3; 95% CI:1.1–2.0); and regular monthly management meetings (aOR = 1.8; 95% CI: 1.0–2.2) were significantly associated with readiness of the integrated services.

In Bangladesh, facilities being managed by the private sector or an NGO (aOR = 3.0; 95% CI:1.7–5.4) compared to the public sector, facilities located in urban locations (aOR = 4.7; 95% CI:2.6–8.3) compared to rural locations, facilities performing routine quality assurance (aOR = 2.3; 95% CI: 1.3–4.0), and having a system to obtain client feedback (aOR = 2.7; 95%

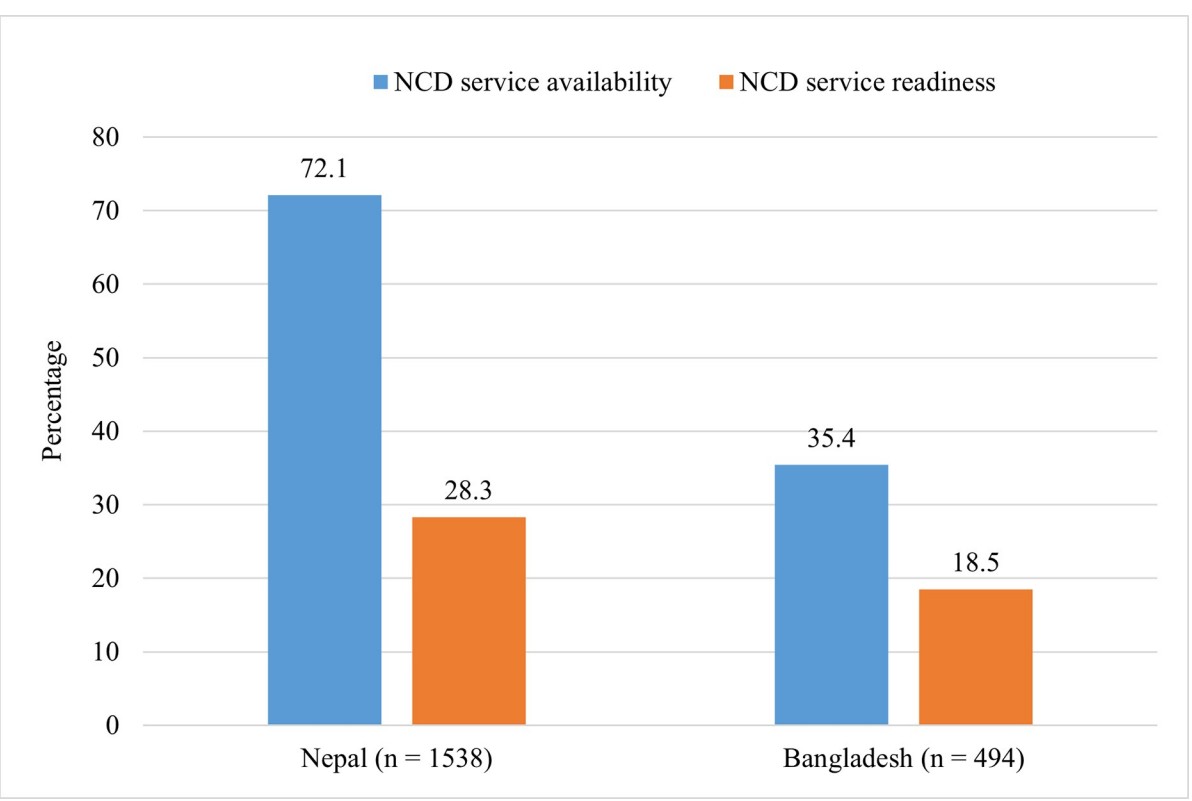

**Fig 1. NCD service availability and readiness among facilities providing ANC service.**

CI:1.4–5.0) were positively associated with readiness for providing ANC and NDC services (see Table 6).

In addition, we assessed whether the facilities that are ready for providing both ANC and NCD services differ from facilities that are ready for ANC only in each country. Most of the factors predicting readiness for only ANC services were similar to those predicting joint readiness in both countries (S1 Table). Health facilities managed by the private sector or an NGO, facilities undertaking routine quality assurance, and having a system to obtain client feedback were positively associated with service readiness for providing only ANC services. Differences were observed in external supervision and regular monthly management meetings. Facilities supervised in the previous 4 months had higher odds for ANC only readiness index, while regular monthly management meetings were not associated with ANC only service readiness in Nepal.

## Discussion

Considering the present and future burden of NCDs, it is imperative that the health care systems in LMICs prioritize strategies to prevent and control NCDs. Women being in close contact with primary health care for maternal health care services during pregnancy can become a bigger part of the solution. This study described service readiness and associated factors for both ANC and NCD services in two LMICs using recent survey data from SPAs.

The findings showed that 71% facilities in Nepal, and 34% in Bangladesh had service availability for both ANC and NCD. Similarly, 24% of facilities in Nepal, and 16% in Bangladesh showed readiness for providing integrated ANC and NCD services. Among the facilities

**Table 6. Factors associated with readiness for providing both ANC and NCD services (binary logistic regression estimating ORs).**

| Variables | Nepal (N = 1565) | | Bangladesh (N = 512) | |
|---|---|---|---|---|
| | Unadjusted OR (95%CI) | Adjusted OR (95%CI) | Unadjusted OR (95%CI) | Adjusted OR (95%CI) |
| **Managing authority** | | | | |
| Public | Ref. | Ref. | Ref. | Ref. |
| Private/NGO | 8.9*** (5.9–13.2) | 10.1*** (6.5–15.7) | 12.6*** (8.1–19.6) | 3.0*** (1.7–5.4) |
| **Location** | | | | |
| Rural | Ref. | Ref. | Ref. | Ref. |
| Urban | 1.3 (0.9–1.8) | 0.9 (0.6–1.3) | 15.4*** (9.5–24.9) | 4.7*** (2.6–8.3) |
| **Routine quality assurance** | | | | |
| Not Performed | Ref. | Ref. | Ref. | Ref. |
| Performed | 1.8** (1.3–2.7) | 1.9** (1.2–2.8) | 3.7*** (2.5–5.5) | 2.3** (1.3–4.0) |
| **System to obtain client feedback** | | | | |
| No | Ref. | Ref. | Ref. | Ref. |
| Yes | 1.7** (1.2–2.5) | 1.3* (1.0–2.0) | 7.2*** (4.5–11.5) | 2.7** (1.4–5.0) |
| **External supervision in the last 4 months** | | | | |
| Did Not Occur | Ref. | Ref. | Ref. | Ref. |
| Occurred | 1.4* (1.0–2.0) | 1.5 (0.9–2.2) | 0.5** (0.3–0.8) | 0.6 (0.3–1.0) |
| **Regular monthly management meetings** | | | | |
| No | Ref. | Ref. | - | - |
| Yes | 1.8** (1.3–2.7) | 1.5* (1.0–2.2) | - | - |

*$P<0.05$

**$P<0.01$

***$P<0.001$, —variables not reported

providing ANC services, NDC service availability was observed in 72% and 35% of facilities in Nepal and Bangladesh respectively, while NCD service readiness was found among 28% of facilities in Nepal and 19% of facilities in Bangladesh. There was a higher proportion of facilities reporting NCD service availability and readiness in Nepal compared to Bangladesh and this may be related to the recent implementation of the Package of Essential Non-communicable Diseases (PEN) at the primary health care level as recommended by the WHO. Additionally, as the survey year was more recent in Nepal (2021 in Nepal and 2017 in Bangladesh), improvement in NCD services might have occurred in more recent times in Bangladesh.

In both countries, a smaller proportion of the health facilities reported having both ANC and NCD guidelines in place, having at least one health provider trained in both ANC and NCD services, and having a laboratory for ANC and NCD services. In terms of domains of ANC readiness, a large proportion of health facilities in both study countries lacked trained staff and guidelines, and diagnostics. In terms of NCD readiness, few facilities had adequate staff and guidelines in place (11.1% in Nepal, and 5.7% in Bangladesh). Further, NCD service readiness of facilities in Nepal and Bangladesh was low in the domains of diagnostics, and medicine and commodities.

There is a substantial gap in the capacity of health facilities to provide both ANC and NCD services in the study countries. In general, gaps were particularly notable in the availability of adequately trained staff and guidelines, basic equipment, diagnostics, and medicines. The presence of guidelines and standard operating procedures coupled with trained staff is essential to providing quality health services. Facilities with clear guidelines are more likely to provide integrated health care services in LMICs [26, 30]. This study suggests that the proportion of health facilities with trained staff and service guidelines for both ANC and NCD was low

across the two study countries. These results are consistent with previous research reporting poor service readiness for NCD in LMICs [29, 31–35].

There was variation in factors associated with the readiness for ANC and NCD services. In general, health facilities managed by the private sector or an NGO, facilities undertaking routine quality assurance, and having a system to obtain client feedback (in both Nepal and Bangladesh) were positively associated with service readiness for providing integrated services. Facilities located in urban areas (in Bangladesh), and facilities undertaking regular management meetings (in Nepal) also had higher odds of readiness index. The findings suggesting poorer readiness of public facilities supported previous research showing better provision of NCD services in private facilities [36–38]. Previous studies have also reported deficits in the training and development of staff, and in equipment and medicine supplies in rural areas [39, 40]. In this study, the analysis of the association between external supervision and joint service readiness yielded somewhat unexpected results: there was no significant association in both Nepal and Bangladesh. Factors predicting readiness for only ANC services are similar to those predicting joint ANC and NCD readiness, with the exception of external supervision (associated with ANC only service readiness) and regular monthly management meetings (did not predict ANC only service readiness).

## Policy implications

Results from this study highlight the challenges for LMICs in providing an integrated ANC and NCD service. Consistent with the literature [10, 41], there is opportunity for LMICs to routinely screen women for common health conditions and NCDs that contribute to pregnancy-related complications, to identify and engage women requiring treatment and preventive care, and to provide ongoing support to both mother and child in the postpartum period and beyond to adopt a healthy lifestyle. This is crucial to prevention of intergenerational disease, and to addressing its long-term health and economic impacts.

Building innovative partnerships and multisectoral collaboration between services to support an integrated approach to better health care is also essential to mothers' and newborns' health, and to their subsequent long-term health outcomes. The analysis of readiness for both ANC and NCD services provides a basis that facilities in LMICs have in providing integrated services. For successful integration of ANC and NCD services, health planners and policymakers should ensure adequate availability of skilled personnel, policy, guidelines and standards, and diagnostics, medicines, and commodities. Evidence provided in this study highlights the gap in joint readiness for integrating ANC and NCD services and the need to strengthen capacity in both Nepal and Bangladesh. There is also a need for policymakers, and service providers to ensure that the health staff are suitably trained to provide person-centred, evidence-based, and culturally competent care. In addition, communities have a vital role to play, and strategies can include training and mobilization of community leaders, community-based organizations, traditional birth attendants, and volunteers, as well as health campaigns, school-based health promotion and home-based care [42].

## Strengths and limitations

To our knowledge, this is the first study to assess the availability and readiness for integration of ANC and NCD services in LMICs dealing with two countries. The study used data from the Demographic and Health Survey Program's SPA, which is a representative and comprehensive nationwide sample survey of health facilities. The outcome variable "readiness for integration" was created based on indicators suggested in the WHO's SARA framework. The provided

estimates were adjusted and weighted to account for cluster sampling, non-response, and disproportionate sampling.

However, the study also had limitations, which should be considered. First, the SARA framework is designed to assess the underlying prerequisites of service quality [34] and the availability and readiness of integrated services; although these are the preconditions for quality care, they do not necessarily indicate that quality, competent care is actually being provided [43]. Second, this study considered only three NCDs–diabetes, cardiovascular diseases, and chronic respiratory diseases. The availability and readiness for services for other NCDs, such as cancer, mental health conditions, and kidney diseases, which are also prevalent in LMICs, may differ. Third, the nationally representative surveys from the two LMICs used in this study may limit the generalizability of the results to other settings. Fourth, although several important health facility- and management-related factors were included in the multivariate logistic regression models, important variables, such as the type of facility and insurance, were not included, as these were not uniformly measured or reported in the included surveys. Further, the lack of similar previous studies assessing facility readiness for ANC and NCD services limits the comparability of these findings with other similar studies.

## Conclusions

Service readiness for integration of ANC and NCD was weak in Nepal and Bangladesh, largely due to shortages in their trained health care workforces, an absence of guidelines and policy, and limited availability of diagnostics, medicines, and commodities. Strengthening the health care system is a priority to facilitate the integration of ANC and NCD services, and as such the above identified issues should be addressed to enable health services to provide integrated care at an acceptable level of quality.

## Supporting information

**S1 Table. Factors associated with readiness for providing only ANC services (binary logistic regression estimating ORs).**
(DOCX)

## Acknowledgments

The authors would like to acknowledge the Demographic and Health Survey Program (DHS) programme for providing access to the dataset.

## Author Contributions

**Conceptualization:** Deependra K. Thapa, Kiran Acharya, Anjalina Karki.

**Data curation:** Kiran Acharya.

**Formal analysis:** Deependra K. Thapa, Kiran Acharya.

**Investigation:** Deependra K. Thapa, Kiran Acharya, Anjalina Karki, Michelle Cleary.

**Methodology:** Deependra K. Thapa, Kiran Acharya, Anjalina Karki.

**Project administration:** Deependra K. Thapa, Anjalina Karki, Michelle Cleary.

**Resources:** Deependra K. Thapa.

**Supervision:** Deependra K. Thapa, Michelle Cleary.

**Validation:** Deependra K. Thapa, Kiran Acharya, Anjalina Karki, Michelle Cleary.

**Visualization:** Deependra K. Thapa, Michelle Cleary.

**Writing – original draft:** Deependra K. Thapa, Anjalina Karki, Michelle Cleary.

**Writing – review & editing:** Deependra K. Thapa, Kiran Acharya, Anjalina Karki, Michelle Cleary.

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
