## [Decision Letter · Decision Letter 0]

27 Apr 2022

PONE-D-21-26082Health facility readiness to provide integrated antenatal care (ANC) and non-communicable disease (NCD) services in Nepal, Bangladesh and Afghanistan: Analysis of facility-based surveysPLOS ONE

Dear Dr. Thapa,

Thank you for submitting your manuscript to PLOS ONE. After careful consideration, we feel that it has merit but does not fully meet PLOS ONE’s publication criteria as it currently stands. Therefore, we invite you to submit a revised version of the manuscript that addresses the points raised during the review process.

The reviewer raised a number of major and minor concerns that must be addressed. They felt that the research question was not presented clearly and therefore the rationale of the study was not fully evident. They also felt that aspects of the methodology were not presented clearly. Their comments can be viewed in full, below.

We look forward to receiving your revised manuscript.

Kind regards,

Natasha McDonald, PhD

Associate Editor

PLOS ONE

Reviewers' comments:

Reviewer's Responses to Questions

**Comments to the Author**

1. Is the manuscript technically sound, and do the data support the conclusions?

Reviewer #1: Partly

2. Has the statistical analysis been performed appropriately and rigorously? 

Reviewer #1: No

3. Have the authors made all data underlying the findings in their manuscript fully available?

Reviewer #1: Yes

4. Is the manuscript presented in an intelligible fashion and written in standard English?

Reviewer #1: Yes

5. Review Comments to the Author

Reviewer #1: Dear Dr. Thapa and colleagues,

I read your article with great interest; the topic area is an important one for further work. I raise below questions and concerns with the current approach.

Major:

1) Conceptual framing. Two issues: what is the value of analyzing multiple countries given the differences in population needs, health system design, and policymaking process? Given for instance the decentralized approach to health service administration in Nepal, what is the value to a district policymaker of seeing data from Nepal compared to Bangladesh and Afghanistan (particularly since, as noted below, the ASPA is quite difficult to compare to any other)? Including multiple countries makes it more difficult to provide a nuanced analysis and policy recommendations specific to one setting, so there should be some clear justification for the value of doing so. Secondly, the concept of integration is tangential to the analysis here, which can speak only to co-occurence of readiness for ANC and NCDs. Readiness for each service does not imply ready to integrate, as the latter requires shared knowledge, space, resources, patient flow, etc. Further, the introduction sets up the analysis more towards the question, 'Of facilities offering ANC, how many might be prepared to provide NCD care if for instance policies focused on NCD services during ANC were pursued?' The analysis itself addresses the question, 'how many facilities meet bare minimum criteria to provide ANC and NCD services?', which is slightly different. I would suggest pulling back the talk of integration and revising the denominator to focus on the more narrow and more actionable question of NCD readiness within ANC facilities. The conceptual basis for the explanatory variables tested in the multivariable regression requires further detail.

2) Methods: inclusion of ASPA. The Afghanistan SPA has a different target population than all other SPA, and as such provides limited insight in a comparative analysis. At most it should be compared only to hospitals in other settings. The description of the ASPA including hospitals "in addition" is misleading - in fact the whole survey is focused on hospitals / referral centers in urban areas.

Minor:

Where did staff training items come from, provider interviews? Operationalized as any respondent indicating relevant training? In results section, it sounds as if the same provider had to be trained in ANC and NCDs, while in the methods section it sounded like this was assessed separately by service (e.g., at least one provider trained in ANC, at least one provider trained in NCDs, whether or not they are the same person). Similarly the laboratory items in SPA are not specific to ANC and NCDs, please clarify

Wide range in CIs in Table 5 suggests sparse data.

Please note if/that sampling weights were used when describing incorporation of survey attributes

What data source was used to check whether Nepal facilities were urban or rural based on the GPS coordinates?

6. PLOS authors have the option to publish the peer review history of their article (what does this mean?). If published, this will include your full peer review and any attached files.

Reviewer #1: No

---

## [Author Response · Author response to Decision Letter 0]

3 Jul 2022

Editor 

The reviewer raised a number of major and minor concerns that must be addressed. They felt that the research question was not presented clearly and therefore the rationale of the study was not fully evident. They also felt that aspects of the methodology were not presented clearly. Their comments can be viewed in full, below. Thank you for the opportunity to revise the manuscript. Comments and suggestions have been addressed as detailed below.

We also thank the reviewers for their positive and encouraging comments and suggestions. These were very helpful and have improved the quality of our paper. We have addressed all comments, amending the manuscript as suggested. 

Reviewer’s Comments:

Major 

1) Conceptual framing. Two issues: what is the value of analyzing multiple countries given the differences in population needs, health system design, and policymaking process? Given for instance the decentralized approach to health service administration in Nepal, what is the value to a district policymaker of seeing data from Nepal compared to Bangladesh and Afghanistan (particularly since, as noted below, the ASPA is quite difficult to compare to any other)? Including multiple countries makes it more difficult to provide a nuanced analysis and policy recommendations specific to one setting, so there should be some clear justification for the value of doing so. Thank you for highlighting this concern. We have amended the manuscript and removed Afghanistan from the analysis. However, both Nepal and Bangladesh have similar health systems, with shared challenges common to low- and middle-income countries. The added value of including these two countries and justification for doing so is now included in the paragraph prior to the Method section of the manuscript.

Secondly, the concept of integration is tangential to the analysis here, which can speak only to co-occurence of readiness for ANC and NCDs. Readiness for each service does not imply ready to integrate, as the latter requires shared knowledge, space, resources, patient flow, etc.

Further, the introduction sets up the analysis more towards the question, 'Of facilities offering ANC, how many might be prepared to provide NCD care if for instance policies focused on NCD services during ANC were pursued?' The analysis itself addresses the question, 'how many facilities meet bare minimum criteria to provide ANC and NCD services?', which is slightly different. I would suggest pulling back the talk of integration and revising the denominator to focus on the more narrow and more actionable question of NCD readiness within ANC facilities. The conceptual basis for the explanatory variables tested in the multivariable regression requires further detail. Thank you for raising this important observation. Our operational definition for service integration is co-occurrence of readiness for ANC and NCDs. In line with the reviewer’s observation, we have re-named the outcome variable from ‘service integration readiness’ to ‘facility readiness to provide both ANC and NCD services’. We have also removed the word ‘integration’ from the title of the manuscript.

The introduction sets up on the opportunity of integrating ANC and NCD services. We aimed to assess readiness for providing ANC and NCD services, which has been made more explicit in our study aim. The introduction section has been revised providing more focus on the importance of readiness to provide both ANC and NCD services. In terms of analysis, previously we have included only those facilities which provided either ANC or NCD services. In the revised version we re-analysed the data including all the health facilities except the stand-alone HIV testing and counselling centres (HTCs) and community clinics which are not expected to provide ANC and/or NCD services. Furthermore, we have also included analysis on facilities showing NCD service readiness among those facilities which were offering any ANC service (Fig 1) (Note: Most of the facilities were providing ANC services).

2) Methods: inclusion of ASPA. The Afghanistan SPA has a different target population than all other SPA, and as such provides limited insight in a comparative analysis. At most it should be compared only to hospitals in other settings. The description of the ASPA including hospitals "in addition" is misleading - in fact the whole survey is focused on hospitals / referral centers in urban areas. We agree and have deleted “in addition” and Afghanistan accordingly. Thank you for raising this. 

Minor 

Where did staff training items come from, provider interviews? Operationalized as any respondent indicating relevant training? In results section, it sounds as if the same provider had to be trained in ANC and NCDs, while in the methods section it sounded like this was assessed separately by service (e.g., at least one provider trained in ANC, at least one provider trained in NCDs, whether or not they are the same person). Similarly the laboratory items in SPA are not specific to ANC and NCDs, please clarify Staff training items were assessed by provider interview, defined as “At least one staff member trained in ANC (for ANC), and diabetes, CVD and CRD diagnosis and treatment (for NCD)”. As we undertook the facility level analysis there could be different health workers trained in ANC or NCDs. 

The diagnostics domain under the ANC service readiness measured services for Hemoglobin (Hb) and Urine (dipstick- protein). Under the NCDs they were Blood glucose, Urine protein, and Urine glucose which were as per the SPA items for service readiness.

Wide range in CIs in Table 5 suggests sparse data. The data has been re-analysed, and addressed the issues identified in the previous version of the manuscript.

Please note if/that sampling weights were used when describing incorporation of survey attributes Sampling weights were used to correct for non-responses and disproportionate sampling. This has been specified in the data analysis section under methodology. 

What data source was used to check whether Nepal facilities were urban or rural based on the GPS coordinates? Location of the facilities (rural and urban) were classified according to the administrative units based on the available GPS coordinates and defined rural (rural municipality) and urban areas (metropolitan/sub metropolitan city and municipality). This has been clarified in the manuscript.

---

## [Decision Letter · Decision Letter 1]

29 Aug 2022

PONE-D-21-26082R1Health facility readiness to provide antenatal care (ANC) and non-communicable disease (NCD) services in Nepal and Bangladesh: Analysis of facility-based surveysPLOS ONE

Dear Dr. Thapa,

Thank you for submitting your manuscript to PLOS ONE. After careful consideration, we feel that it has merit but does not fully meet PLOS ONE’s publication criteria as it currently stands. Therefore, we invite you to submit a revised version of the manuscript that addresses the points raised during the review process. As noted by the review below, the findings are primarily driven by NCD services given the broad availability of ANC; further consideration of what distinguishes the facilities offering and ready to provide both services compared to one or neither is necessary to strengthen the manuscript. 

We look forward to receiving your revised manuscript.

Kind regards,

Hannah Hogan Leslie, PhD

Guest Editor

PLOS ONE

Additional Editor Comments:

Please note the reviewer's suggestion of comparing facilities offering ANC and NCD services to those who can offer neither or only one as a way of providing greater insight on what determines joint readiness. The reviewer aptly notes that the current findings are not substantially different from availability and readiness of NCD services alone given the broad availability of ANC services. Tabular presentation of this 4-group comparison would enrich the study and provide more substance to this work; the comparison could be repeated considering readiness as well as availability.

Reviewers' comments:

Reviewer's Responses to Questions

**Comments to the Author**

1. If the authors have adequately addressed your comments raised in a previous round of review and you feel that this manuscript is now acceptable for publication, you may indicate that here to bypass the “Comments to the Author” section, enter your conflict of interest statement in the “Confidential to Editor” section, and submit your "Accept" recommendation.

Reviewer #2: (No Response)

2. Is the manuscript technically sound, and do the data support the conclusions?

Reviewer #2: (No Response)

3. Has the statistical analysis been performed appropriately and rigorously? 

Reviewer #2: (No Response)

4. Have the authors made all data underlying the findings in their manuscript fully available?

Reviewer #2: (No Response)

5. Is the manuscript presented in an intelligible fashion and written in standard English?

Reviewer #2: (No Response)

6. Review Comments to the Author

Reviewer #2: Thank you for the opportunity to review this manuscript. It investigates a topic of importance: to what extent are health facilities in LMICs (here, Bangladesh and Nepal) ready to offer NCD services for women seeking ANC. I do not feel the analysis offers an advance beyond what we already know, namely that very few facilities offer and/or are ready to provide NCD services. The “combined” availability and readiness numbers are presumably heavily driven by the NCD numbers (they parallel them quite closely. To my understanding the authors did not look at “types” of facilities, i.e. can offer ANC only, can offer NCD only, can offer both, can offer neither – this possibly could have highlighted something new if, for example, the facilities offering ANC+NCD are a unique subset of those offering NCD-only. These data have been analyzed in numerous papers making a similar point about NCD services, and the data are quite old at this point (5 and 7 years). The paper is well-written and clear but I just do not feel that it offers anything that has not already been well-documented previously.

7. PLOS authors have the option to publish the peer review history of their article (what does this mean?). If published, this will include your full peer review and any attached files.

Reviewer #2: No

---

## [Author Response · Author response to Decision Letter 1]

9 Nov 2022

Comment: Thank you for the opportunity to review this manuscript. It investigates a topic of importance: to what extent are health facilities in LMICs (here, Bangladesh and Nepal) ready to offer NCD services for women seeking ANC. 

Response: Many thanks for your comments. This paper assesses readiness of health facilities to provide both ANC and NCD services.

We have addressed comments as below.

Comment: I do not feel the analysis offers an advance beyond what we already know, namely that very few facilities offer and/or are ready to provide NCD services. The “combined” availability and readiness numbers are presumably heavily driven by the NCD numbers (they parallel them quite closely).

Response: Thank you for highlighting this concern. We have updated 2015 data for Nepal with the recently released 2021 health facility survey. The results shows that 72% of health facilities in Nepal (2021) and 35% in Bangladesh (2017) have availability of NCD service, while the NCD readiness is 28.4% and 18.2% respectively in these countries. 

Comment: To my understanding the authors did not look at “types” of facilities, i.e. can offer ANC only, can offer NCD only, can offer both, can offer neither – this possibly could have highlighted something new if, for example, the facilities offering ANC+NCD are a unique subset of those offering NCD-only.

Response: Many thanks for this suggestion. We have added 4-group comparison (only ANC, only NCD, both ANC and NCD, and neither) for availability, and readiness.

Comment: These data have been analyzed in numerous papers making a similar point about NCD services, and the data are quite old at this point (5 and 7 years). The paper is well-written and clear but I just do not feel that it offers anything that has not already been well-documented previously.

Response: We have replaced the 2015 data from Nepal with the recently released 2021 Health Facility Survey in support of the contribution this paper we believe makes.

---

## [Editor Report · Decision Letter 2]

6 Dec 2022

PONE-D-21-26082R2Health facility readiness to provide antenatal care (ANC) and non-communicable disease (NCD) services in Nepal and Bangladesh: Analysis of facility-based surveysPLOS ONE

Dear Dr. Thapa,

Thank you for submitting your manuscript to PLOS ONE. After careful consideration, we feel that it has merit but does not fully meet PLOS ONE’s publication criteria as it currently stands. Therefore, we invite you to submit a revised version of the manuscript that addresses the points raised during the review process. Please ensure that your decision is justified on PLOS ONE’s publication criteria and not, for example, on novelty or perceived impact.

 Please submit your revised manuscript by Jan 20 2023 11:59PM. If you will need more time than this to complete your revisions, please reply to this message or contact the journal office at plosone@plos.org. Please include the following items when submitting your revised manuscript:A rebuttal letter that responds to each point raised by the academic editor and reviewer(s). You should upload this letter as a separate file labeled 'Response to Reviewers'.A marked-up copy of your manuscript that highlights changes made to the original version. You should upload this as a separate file labeled 'Revised Manuscript with Track Changes'.An unmarked version of your revised paper without tracked changes. You should upload this as a separate file labeled 'Manuscript'.If applicable, we recommend that you deposit your laboratory protocols in protocols.io to enhance the reproducibility of your results. Protocols.io assigns your protocol its own identifier (DOI) so that it can be cited independently in the future. For instructions see: https://journals.plos.org/plosone/s/submission-guidelines#loc-laboratory-protocols. Additionally, PLOS ONE offers an option for publishing peer-reviewed Lab Protocol articles, which describe protocols hosted on protocols.io. Read more information on sharing protocols at https://plos.org/protocols?utm_medium=editorial-email&utm_source=authorletters&utm_campaign=protocols.

We look forward to receiving your revised manuscript.

Kind regards,

Hannah Hogan Leslie, PhD

Guest Editor

PLOS ONE

Journal Requirements:

Additional Editor Comments (if provided):

I appreciate the careful revision and the decision to update the article to reflect the latest Nepal data. I would make several minor suggestions:

- revise wording of 'willingness to provide' as 'capacity' since readiness is about tangible capacity, not provider motivation / interest in providing a service

- replace 'weightage' with weight or weighting

- remove 'integration' from results section wording (consider 'joint readiness') and revisit in discussion how the analysis of readiness for both ANC and NCD speaks to the basis that facilities have for providing integrated services.

It would be interesting to see if the facilities that are ready for both services differ from facilities ready for ANC only in each country, and if the factors predicting this joint vs. single readiness are similar to those distinguishing joint readiness from all other facilities. This is to the authors' consideration, and not a requirement for further consideration.
---

## [Author Response · Author response to Decision Letter 2]

29 Dec 2022

Editor: I appreciate the careful revision and the decision to update the article to reflect the latest Nepal data. I would make several minor suggestions:

Response: Thank you for your supportive comments, and the opportunity to revise the manuscript. 

Editor: We have addressed the comments as detailed below.

- revise wording of 'willingness to provide' as 'capacity' since readiness is about tangible capacity, not provider motivation / interest in providing a service

Response: Revised as suggested.

Editor: - replace 'weightage' with weight or weighting

Response: Thank you, we have changed ‘weightage’ to ‘weighting’ as suggested. 

Editor: - remove 'integration' from results section wording (consider 'joint readiness') and revisit in discussion how the analysis of readiness for both ANC and NCD speaks to the basis that facilities have for providing integrated services.

Response: The word ‘integration’ in the Results section has been replaced with ‘joint readiness’. We have revised the Discussion section (on policy implications) and included how the analysis of readiness for both ANC and NCD speaks to the basis that facilities have for providing integrated services; as below:

The analysis of readiness for both ANC and NCD services provides a basis that facilities in LMICs have in providing integrated services. For successful integration of ANC and NCD services, health planners and policymakers should ensure adequate availability of skilled personnel, policy, guidelines and standards, and diagnostics, medicines, and commodities. Evidence provided in this study highlights the gap in joint readiness for integrating ANC and NCD services and the need to strengthen capacity in both Nepal and Bangladesh. .

Editor: It would be interesting to see if the facilities that are ready for both services differ from facilities ready for ANC only in each country, and if the factors predicting this joint vs. single readiness are similar to those distinguishing joint readiness from all other facilities. This is to the authors' consideration, and not a requirement for further consideration. Thanks for this comment.

Response: We analyzed whether the facilities that are ready for providing both ANC and NCD services differ from facilities ready for ANC only in each country. The findings are presented in the Supplementary file 1 and are also described in the Results section. As below:

In addition, we assessed whether the facilities that are ready for providing both ANC and NCD services differ from facilities that are ready for ANC only in each country. Most of the factors predicting readiness for only ANC services were similar to those predicting joint readiness in both countries (Supplementary file 1). Health facilities managed by the private sector or an NGO, facilities undertaking routine quality assurance, and having a system to obtain client feedback were positively associated with service readiness for providing only ANC services. Differences were observed in external supervision and regular monthly management meetings. Facilities supervised in the previous 4 months had higher odds for ANC only readiness index, while regular monthly management meetings were not associated with ANC only service readiness in Nepal.

Reviewer’s Comments: None Thank you. No action required.

---

## [Editor Report · Decision Letter 3]

23 Jan 2023

Health facility readiness to provide antenatal care (ANC) and non-communicable disease (NCD) services in Nepal and Bangladesh: Analysis of facility-based surveys

PONE-D-21-26082R3

Dear Dr. Thapa,

We’re pleased to inform you that your manuscript has been judged scientifically suitable for publication and will be formally accepted for publication once it meets all outstanding technical requirements. Thank you for your assiduous efforts to improve this work and ensure it is suitable for publication.

Kind regards,

Hannah Hogan Leslie, PhD

Guest Editor

PLOS ONE